# EFFICIENT VISION-LANGUAGE MODELS BY SUMMARIZING VISUAL TOKENS INTO COMPACT REGISTERS

## ABSTRACT

Recent advancements in vision-language models have expanded their potential for real-world applications, enabling these models to perform complex reasoning on images. However, in the widely used fully autoregressive pipeline like LLaVA, where projected visual tokens are prepended to textual tokens, the visual tokens often number in the hundreds or thousands, making them much longer than the input prompt. This large quantity of visual tokens introduces significant computational overhead, slowing down training and inference. In this paper, we propose **Vi**sual **C**ompact **To**ken **R**egisters (`Victor`), a method that reduces the number of visual tokens by summarizing them into a smaller set of register tokens. `Victor` adds a few learnable register tokens after the visual tokens and summarizes the visual information into these registers using the first few layers in the language tower. After these few layers, all visual tokens are discarded, significantly improving computational efficiency for both training and inference. Notably, our method is easy to implement and requires a small number of new trainable parameters with minimal impact on model performance. In our experiment, with merely 8 visual registers—about $1\%$ of the original tokens—`Victor` shows less than a $4\%$ performance drop while reducing total training time by $43\%$ and boosting inference throughput by $3.36\times$.

## 1 INTRODUCTION

Vision-language models (VLMs) have attracted considerable attention for their capability to process visual and textual information, enabling various real-world applications such as image captioning, visual question answering, and multimodal reasoning (OpenAI, 2023; Liu et al., 2024c). For example, GPT-4V (OpenAI, 2023) demonstrates the potential of these models in helping visually impaired individuals to "see" the world through cell phone cameras.

Recent vision-language models, such as LLaVA (Liu et al., 2024c), employ a pre-trained vision encoder as the model's "eye" to extract visual features and use a pre-trained language model as the "brain" to perform reasoning and text generation. This straightforward architecture is highly effective and requires only a small instruction dataset for fine-tuning. However, since the entire set of projected image features is fed as the input to the language model, it results in a significant computational overhead due to the large number of visual tokens. For instance, LLaVA-NeXT (Liu et al., 2024b) utilizes $2,880$ tokens to represent a single image, which can be overly redundant in many scenarios. In contrast, the average text instruction length across all benchmarks used in LLaVA-NeXT is fewer than 70 tokens, as shown in Appendix A.1.

Therefore, to improve the efficiency of vision-language models, it is clear that reducing the number of visual tokens is essential. The state-of-the-art method, FastV (Chen et al., 2024), achieves this by directly dropping unimportant visual tokens. This approach is highly effective when reducing the number of tokens by up to half. However, the model's performance drops significantly when more than half of the tokens are removed. Meanwhile, FastV requires obtaining the attention scores from the self-attention block. Efficient attention implementations, such as FlashAttention (Dao et al., 2022; Dao, 2023), do not support this feature. Consequently, FastV relies on standard attention implementations to retrieve the attention scores. This can limit its deployment on devices that do not support such implementations. We also find that this constraint slows down the fine-tuning stage, as presented in Section 5.3.

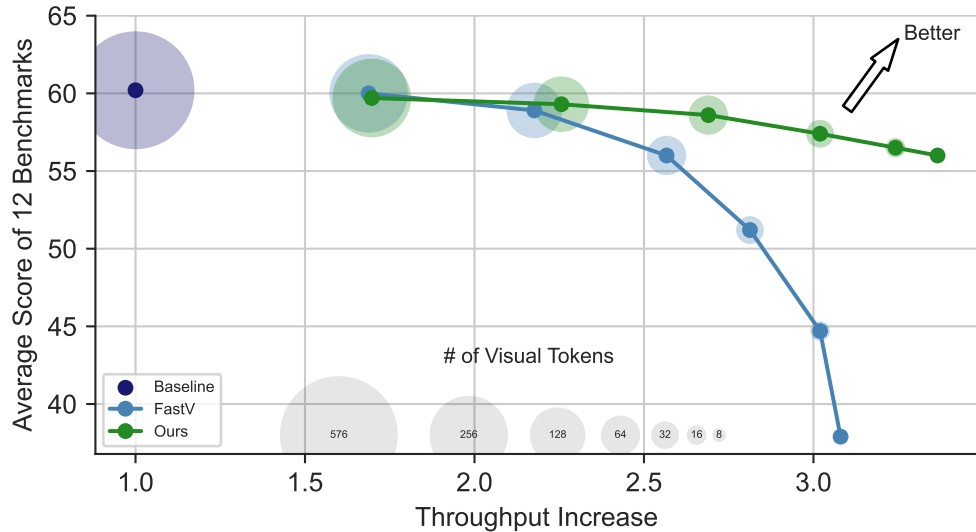

Figure 1: **Efficiency-Performance Trade-Off Curve.** We compare our proposed method, `Victor`, with the state-of-the-art method FastV. The normalized average score across 12 benchmarks and the corresponding throughput increase relative to the original baseline model are reported (details in Section 4.2). The size of the circles indicates the number of visual tokens for each method, with larger circles representing more tokens. `Victor` establishes a more favorable Pareto frontier than FastV, demonstrating a significantly smaller performance drop as throughput increases.

On the other hand, another popular approach for token reduction is to use a transformer-based projection model for condensing visual tokens into a smaller set of queries. Notable examples include the Perceiver Resampler (Jaegle et al., 2021; Alayrac et al., 2022; Bai et al., 2023) and Q-Former (Li et al., 2023b). These methods outperform FastV when the reduced set of visual tokens is much smaller than the original. However, they require much more trainable parameters.

In this paper, we introduce **Vi**sual **C**ompact **To**ken **R**egisters (`Victor`), a simple yet effective early visual token summarizing method that provides a superior efficiency-quality trade-off compared to the state-of-the-art techniques. For visual tokens, we observe that they often contain redundant information, with many tokens being highly similar, as discussed in Section 3.1. To address this, our approach leverages the language model to summarize these visual tokens into compact registers. `Victor` begins by appending a small subset of register tokens to the visual tokens and uses the initial $k$ layers of the language model to summarize visual information into these register tokens. Notably, during training, no specific loss function or operation is applied to explicitly force the visual information into the registers. Empirical results show that the language model naturally uses these tokens to store visual information. Meanwhile, starting at layer $k$, all visual tokens are discarded, leaving only the summarized registers and textual tokens for efficient inference in the subsequent layers.

Our method does not rely on attention scores, making it compatible with the most efficient attention implementations. Moreover, `Victor` introduces just 1.78M additional parameters, accounting for only 0.03% of the total model size. In contrast to approaches like the Perceiver Resampler, which incorporates a separate transformer with 252.86M parameters (3.61% of the total model), `Victor` leverages the powerful language model itself to perform this task, achieving significantly better performance.

As illustrated in Figure 1, `Victor` establishes a more favorable Pareto frontier than the state-of-the-art method, particularly exhibiting a noticeably smaller performance drop as throughput increases. For instance, when the number of visual registers is set to 8—approximately only 1% of the original visual tokens—our model experiences a performance drop of less than 4% while reducing total training time by 43% and achieving a 3.36× increase in inference throughput. Our extensive experiments demonstrate the effectiveness of `Victor` in balancing both efficiency and performance, making it a promising solution for visual token reduction in vision-language models.

## 2 RELATED WORK

### 2.1 VISION-LANGUAGE MODELS

Modern vision-language models typically combine a pre-trained image encoder with a large language model to handle multimodal data (Li et al., 2023b). One popular approach, often referred to as the autoregressive or LLaVA-style model (Li et al., 2023b; Liu et al., 2024c), directly projects visual features into the input embedding space of the language model, treating these features as part of the input tokens. However, in this design, the number of visual tokens is large and often exceeds the number of textual tokens, leading to inefficiencies. Another common approach is cross-attention-based fusion (Alayrac et al., 2022), where added cross-attention blocks inside of the language transformer layers allow textual tokens to attend to visual tokens. More recently, early-fusion models like Fuyu (Bavishi et al., 2023), MoMA (Lin et al., 2024), and Chameleon (Team, 2024) use a unified transformer that processes raw textual tokens and visual patches simultaneously. Additionally, a key component of modern vision-language models recipe is instruction fine-tuning (Dai et al., 2023; Zhu et al., 2023; Liu et al., 2024a;c; Singla et al., 2024), which enables the model to function as a typical chatbot while also processing images, even with a small synthetic fine-tuning dataset. In this paper, we focus on LLaVA-style models.

### 2.2 VISUAL TOKEN REDUCTION

To improve the efficiency of vision or vision-language models, pruning or distilling visual tokens has been widely studied. Rao et al. (2021) introduce DynamicViT, which uses a small module to predict the importance of each visual token, dropping less important ones to enhance efficiency. Similarly, EViT (Liang et al., 2022) retains important tokens and fuses the less important ones within the model, using attention scores from the class token to the visual tokens. Further, PuMer (Cao et al., 2023) reduces the number of both textual and visual tokens by progressively pruning and merging them throughout the cross-modal encoder. Another interesting approach by Saifullah et al. (2023) involves discretizing visual features into textual tokens to reduce dimensionality. For more recent vision-language models, Perceiver Resampler (Jaegle et al., 2021; Alayrac et al., 2022; Bai et al., 2023) and Q-Former (Li et al., 2023b) are commonly used to pool visual tokens into a smaller set of queries using a transformer-based model. Additionally, in the FastV paper (Chen et al., 2024), the authors observe that in LLaVA-style models, the attention from textual tokens to visual tokens significantly diminishes after the first few layers, with the attentiveness declining close to zero after 10% of the transformer layers. Intuitively, their proposed state-of-the-art method drops the unimportant visual tokens accordingly after the first few layers.

### 2.3 VISUAL REGISTERS

Burtsev et al. (2020) first introduce memory tokens, which function similarly to registers. These tokens are used to store global information, enabling the model to effectively handle long-context tasks. Darcet et al. (2023) apply the idea of registers to ViTs. In their work, the authors observe that vision transformer models implicitly use low-information tokens to store global information for internal computations. However, this leads to abnormally high norms for these tokens, making it difficult to interpret the attention maps. Therefore, to address this, they introduce additional register tokens at the end of the sequence to handle this task. This approach not only improves the interpretability of attention maps but also boosts model performance. In our work, we show that these register tokens also enhance the performance of vision-language models, as demonstrated in Section 5.6. However, our primary focus in this paper is on using these registers for information distillation, enabling the model to condense visual information into the registers to improve efficiency.

## 3 METHOD

### 3.1 MOTIVATION

We begin with a key observation: many visual tokens exhibit significant redundancy. As illustrated in Figure 3, the cosine similarities between baseline visual tokens cluster around 1, indicating a high degree of token similarity. This suggests that compressing the visual tokens into a smaller set would result in minimal information loss. To achieve this, we append a set of learnable register tokens to the visual tokens and leverage the language model to summarize the visual information

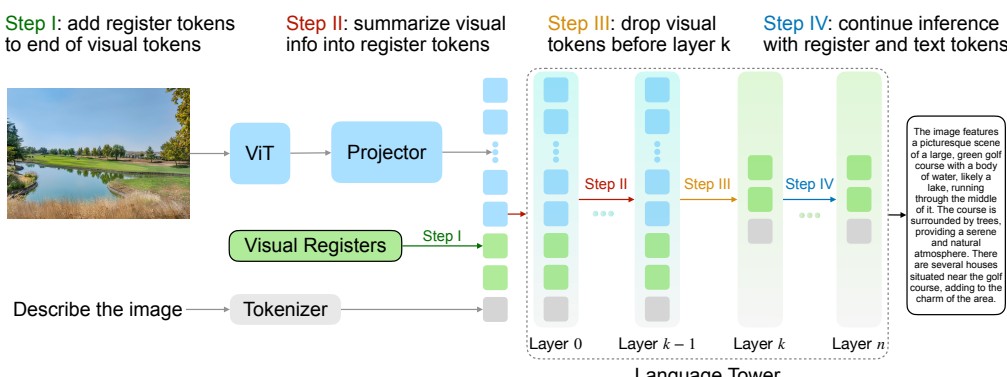

Figure 2: **Method Overview.** `Victor` is a simple yet effective method for enhancing the efficiency of vision-language models. The process involves four key steps based on the LLaVA-style model: (I) appending learned visual register tokens after the visual tokens, where the number of visual registers is much smaller than the number of the visual tokens, (II) using the first $k$ layers of the language tower to summarize visual information into the visual registers, (III) discarding all visual tokens before layer $k$, and (IV) starting from layer $k$, the model performs efficient inference using only the visual registers and textual tokens with significantly reduced sequence length.

into these registers. Rather than using a separate model like the Perceiver Resampler, we utilize the more powerful language model for this task, as it inherently understands which visual tokens are important and how to organize the information. Consequently, as demonstrated in Figure 3, our compact visual registers show reduced redundancy compared to the baseline visual tokens.

Furthermore, based on observations from FastV (Chen et al., 2024), textual tokens in the language model primarily attend to visual tokens in the early transformer layers, with attention scores to visual tokens dropping to nearly zero after the initial layers. This suggests the language model requires only a few layers to process the visual information. Thus, instead of using the entire model, we employ only the first few transformer layers to summarize the visual tokens into the register tokens. After summarization, the visual tokens are discarded, improving model efficiency. An overview of our method is provided in Figure 2[1].

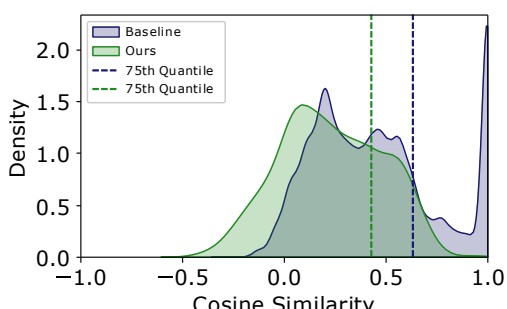

Figure 3: **Token Similarities.**

### 3.2 VICTOR

We now formally introduce our method: `Victor` (**Vi**sual **C**ompact **To**ken **R**egisters). A LLaVA-style vision-language model consists of three main components: (1) the image tower $\mathcal{I}$, which is a pre-trained vision model, such as the CLIP image encoder (Radford et al., 2021); (2) the language tower $\mathcal{T}$, a pre-trained LLM, such as LLaMA (Touvron et al., 2023); and (3) a projector $\mathcal{P}$ that bridges the two, mapping image features into the input embedding space of the language model. Given an image $x_{\text{img}}$, we first extract its features using the image tower $\mathcal{I}$ and produce a set of projected visual tokens $x_V = \{x_V^0, x_V^1, \ldots, x_V^{N-1}\}$ from the projector $\mathcal{P}$.

As described in Algorithm 1, for `Victor`, we additionally introduce a set of learnable visual registers $x_R = \{x_R^0, x_R^1, \ldots, x_R^{M-1}\}$, where $M$ is a hyperparameter controlling the number of visual registers. A smaller $M$ results in a more efficient model, and usually $M \ll N$. We then concatenate the projected visual tokens, visual registers, and textual tokens to form the input: $x = [x_V; x_R; x_T]$.

---

[1]Image credit: https://unsplash.com/photos/green-grass-field-near-lake-under-blue-sky-during-daytime-d3Jf3avtXSg

---

**Algorithm 1** `Victor` - **Vi**sual **C**ompact **To**ken **R**egisters

---

1: **input** Projected Visual Tokens: $x_V = \{x_V^0, x_V^1, \ldots, x_V^{N-1}\}$, Textual Input Tokens: $x_T = \{x_T^0, x_T^1, \ldots, x_T^{L-1}\}$, Visual Registers: $x_R = \{x_R^0, x_R^1, \ldots, x_R^{M-1}\}$, Language Tower: $\mathcal{T}$, Number of Layers: $n$, Drop Layer Index: $k$

2: Form input $x = [x_V; x_R; x_T]$         ▷ Add visual registers to end of visual tokens

3: **for** each layer $i = 0$ to $n$ **do**

4:      **if** $i == k$ **then**

5:         $x = x[N :]$                    ▷ Drop visual tokens before layer $k$

6:      $x = \mathcal{T}_i(x)$

7: **return** $x$

---

This input is processed through the language tower for the first $k$ layers. At the start of layer $k$, all visual tokens are discarded, and the model continues with the truncated hidden states for the remaining layers.

During training, we do not explicitly force the language model to summarize the visual information into the visual registers, but we empirically observe that it does so implicitly. In Section 5.7, we provide an empirical analysis showing that the visual registers effectively summarize important information from the visual tokens. Moreover, because `Victor` leverages the language model itself for this summarization, rather than relying on an external model, and the language model is both powerful and knowing at identifying the most useful image features, our method experiences minimal performance drop compared to approaches like Perceiver Resampler while requiring much fewer additional model parameters.

In practice, we typically set $k$ to 3, which is just roughly 10% of the language tower. This means the language tower processes the full-length hidden states for only the first 10% of its layers. For the remaining 90% layers, it operates on significantly shorter hidden states, thereby improving model efficiency. FastV (Chen et al., 2024), a state-of-the-art method, follows a similar approach by dropping unimportant tokens in the early layers and achieves a comparable theoretical reduction in FLOPs. However, we find that because FastV relies on attention scores to determine which tokens to drop, it cannot utilize efficient attention implementations like FlashAttention (Dao et al., 2022; Dao, 2023) or PyTorch High-Performance Scaled Dot Product Attention (SDPA). Consequently, FastV delivers less throughput improvement than `Victor` when applying the same token-drop ratio in practical scenarios, and also empirically experiences a greater performance degradation in high token-drop ratio regimes.

## 4 EXPERIMENTS

### 4.1 EXPERIMENTAL SETUP

In this paper, we primarily follow the setting of the open-sourced LLaVA-v1.5 (Liu et al., 2024a). The training consists of two main stages: pre-training and instruction fine-tuning.

**Pre-trained Models.** For the image tower, we use the pre-trained OpenAI CLIP ViT-Large model (Radford et al., 2021), and for the text tower, we use the Vicuna-7B-v1.5 model (Zheng et al., 2024), an instruction fine-tuned version of LLaMA-2-7B (Touvron et al., 2023). We also show the results using various language towers in Section 5.4, including Vicuna-13B-v1.5 (Zheng et al., 2024), Meta-Llama-3-8B-Instruct (Dubey et al., 2024), Mistral-7B-Instruct-v0.2 (Jiang et al., 2023), and Qwen2-7B-Instruct (Yang et al., 2024).

**Datasets.** For pre-training, we use the LLaVA CC3M pre-training dataset (Liu et al., 2024c)[2], a subset of 595K images from the CC3M dataset (Sharma et al., 2018). For instruction fine-tuning, we use LLaVA-v1.5-mix665K[3], a mixed dataset comprising COCO 2017 (Lin et al., 2014), GQA

---

[2] https://huggingface.co/datasets/liuhaotian/LLaVA-CC3M-Pretrain-595K
[3] https://huggingface.co/datasets/liuhaotian/LLaVA-Instruct-150K/blob/main/llava_v1_5_mix665k.json

(Hudson & Manning, 2019), OCR-VQA (Mishra et al., 2019), TextVQA (Singh et al., 2019), and VisualGenome (Krishna et al., 2017).

**Implementation.** We follow the hyperparameters used by Liu et al. (2024c). During pre-training, we freeze both the image and text towers, training only the projector and registers. We use the AdamW optimizer (Loshchilov & Hutter, 2017) with a learning rate of 0.0001 and no weight decay for one epoch. In the fine-tuning stage, we unfreeze the text tower while keeping the image tower frozen. Similar to pre-training, we use the AdamW optimizer with a learning rate of 0.00002 and no weight decay for one epoch.

## 4.2 EVALUATION

For evaluation, we use 11 benchmarks from the LLaVA-v1.5 report (Liu et al., 2024a), supplemented by MMMU (Yue et al., 2024), a widely-used benchmark for assessing modern vision-language models. The benchmarks include: VQAv2 (Goyal et al., 2017), GQA (Hudson & Manning, 2019), ScienceQA (Lu et al., 2022), TextVQA (Singh et al., 2019), VizWiz-VQA (Gurari et al., 2018), POPE (Li et al., 2023c), MME-P (Yin et al., 2023), MMBench (Liu et al., 2023), SEED-Bench (Li et al., 2023a), LLaVA-Bench-in-the-Wild (Liu et al., 2024c), MM-Vet (Yu et al., 2023), and MMMU (Yue et al., 2024). These benchmarks provide a comprehensive assessment of models' multi-modal reasoning capabilities, encompassing both academic-task-oriented and instruction-following tasks. We employ the LMMs-Eval framework[4] (Zhang et al., 2024) for evaluation. For simplicity, we primarily report the average of the normalized benchmark scores. Specifically, for MME-P, the score is divided by $2,000$, which represents the full score, as their metric is calculated by summing the accuracies of individual subtasks.

We evaluate efficiency by measuring the increase in throughput with the KV-cache on (Pope et al., 2023). After gathering statistics from all 12 benchmarks, presented in Appendix A.1, we evaluate two settings: 1) 2-token generation and 2) 128-token generation. The 2-token generation simulates scenarios where questions expect a single word, as in GQA (Hudson & Manning, 2019) and TextVQA (Singh et al., 2019). In contrast, the 128-token generation represents open-ended question scenarios, such as in LLaVA-Bench-in-the-Wild (Liu et al., 2024c) and MM-Vet (Yu et al., 2023). In both settings, we use a text prompt length of 64 and a batch size of 16. We choose a batch size of 16 because it is the largest batch size that fits in memory. All training is conducted on 8 NVIDIA A100 GPUs, with efficiency profiling performed on a single NVIDIA A100. By default, we set $k$ to 3 and vary the number of final visual tokens to 256, 128, 64, 32, 16, and 8 for all methods to thoroughly assess the trade-off between efficiency and performance, where the number of visual tokens for the original LLaVA-v1.5 model is 576.

We also compare our methods against two baselines: FastV (Chen et al., 2024) and Perceiver Resampler (Jaegle et al., 2021). FastV is the state-of-the-art token reduction method that filters out less important vision tokens based on attention scores, while the Perceiver Resampler is a compact, transformer-based model designed to condense input tokens into a smaller query set.

## 5 RESULTS

## 5.1 THROUGHPUT INCREASE

We present the efficiency and performance trade-offs for both generation settings in Figure 4, while the performance on individual benchmarks is included in Appendix A.2. As shown, our method has a better Pareto frontier than FastV and the Perceiver Resampler in both scenarios.

Both `Victor` and FastV maintain minimal performance degradation when the throughput increases by approximately $1.5\times$ to $2\times$ and the number of tokens decreases from 576 to 256 or 128. However, FastV's performance declines rapidly beyond this point. In contrast, our method exhibits only a $4\%$ performance drop even when the number of visual tokens is reduced to 8, which is roughly $1\%$ of the original visual token count. Additionally, given the same number of final visual tokens, our method has slightly higher theoretical FLOPs than FastV due to the inclusion of extra register tokens in the initial layers. However, in practice, our method achieves a greater increase in throughput

---

[4] https://github.com/EvolvingLMMs-Lab/lmms-eval

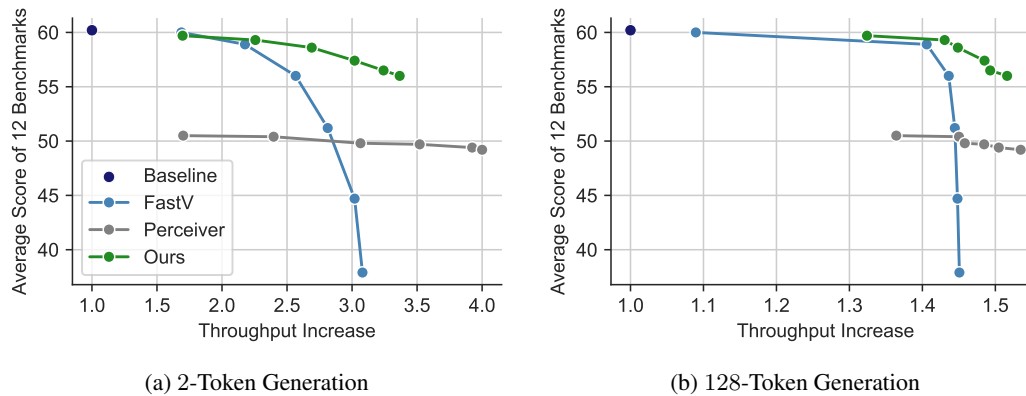

(a) 2-Token Generation          (b) 128-Token Generation

Figure 4: **Efficiency-Performance Trade-Off Curve.** We measure the relative throughput increase compared to the baseline model. The test covers two main scenarios: generating 2 tokens and generating 128 tokens. In both cases, the batch size is set to 16, and the text prompt length is 64 tokens. For all methods, we use 256, 128, 64, 32, 16, and 8 visual tokens to generate the line plot.

compared to FastV. This is due to the fact that, in the layer where FastV performs filtering, the model is constrained to using the original attention mechanism to compute attention scores, as it cannot leverage more efficient attention implementations. In contrast, our method is compatible with a wide range of efficient attention implementations including those that do not support returning attention scores. As a result, `Victor` not only achieves better throughput and more effectively retains the accuracy but is also more adaptable across different devices than FastV.

In contrast, the Perceiver Resampler experiences a substantial performance drop of approximately $10\%$ compared to the original model, performing significantly worse than `Victor`. Interestingly, its performance remains stable across different reduction ratios, consistent with the findings of Laurençon et al. (2024). Despite this performance decline, the Perceiver Resampler achieves a much higher throughput increase than FastV and `Victor`. As shown in Table 1, however, the Perceiver Resampler requires a substantially larger number of additional parameters—252.86M, representing $3.61\%$

Table 1: **Number of Extra Parameters for Different Methods.** The final number of visual tokens is 256.

| Method | # of Extra Parameters |
|---|---|
| FastV | 0 (0.00%) |
| Perceiver | 252.86M (3.61%) |
| Ours | 1.78M (0.03%) |

of the total model—while our method adds only 1.78M, approximately $0.03\%$ of the entire model.

## 5.2 FLOPs REDUCTION

We also report the theoretical FLOPs reduction of the methods, calculated using the FLOPs formula from Chen et al. (2024). As demonstrated in Figure 5, while our method shows a slightly smaller FLOPs reduction due to the presence of additional register tokens at the start of the language tower, the overall reduction is comparable under the significantly higher token reduction rate. Although the Perceiver Resampler achieves a notable increase in throughput, its FLOPs reduction is substantially lower than that of FastV and `Victor`, primarily due to the additional transformer layers it employs.

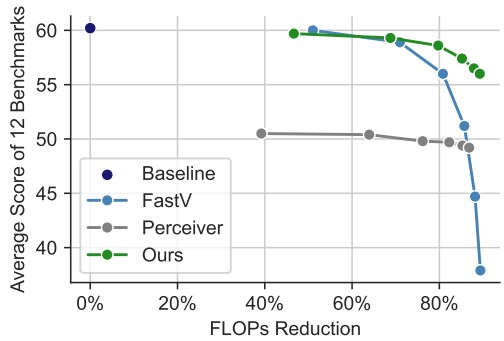

Figure 5: **Performance vs. FLOPs Reduction.**

## 5.3 TRAINING-TIME REDUCTION

`Victor` not only reduces inference costs but also lowers training costs. As indicated in Figure 6, both Perceiver Resampler and `Victor` significantly reduce training time in both pre-training and

fine-tuning stages, with the reduction being especially notable during pre-training due to the shorter text tokens. Victor achieves a greater overall time reduction. In contrast, training with FastV only reduces pre-training time and does not improve fine-tuning efficiency. This is because fine-tuning typically involves a large number of text tokens (often exceeding a thousand), and the use of a naive attention implementation in this phase introduces significant overhead, reducing training efficiency. Additionally, we observe that training with FastV does not match the performance of inference-time FastV. However, it exhibits slower benchmark performance decay as the number of visual tokens decreases and outperforms inference-time FastV when the number of visual tokens drops below 32.

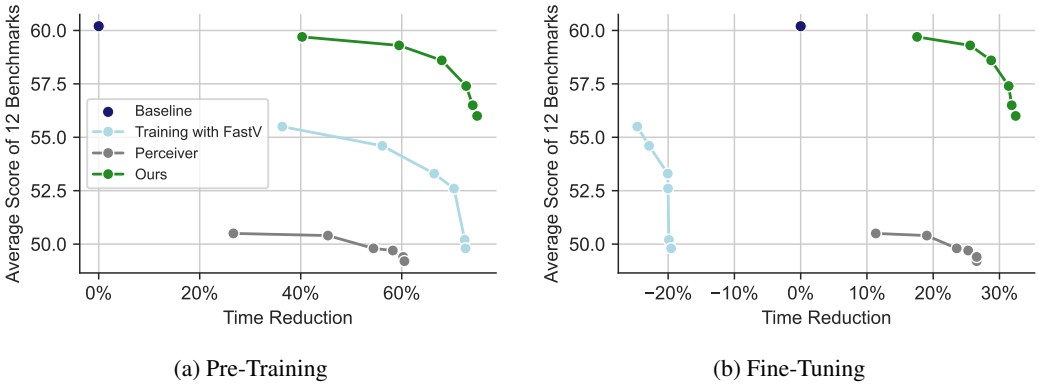

(a) Pre-Training                   (b) Fine-Tuning

Figure 6: **Performance vs. Training-Time Reduction.** We show total training-time reduction in Appendix A.3.

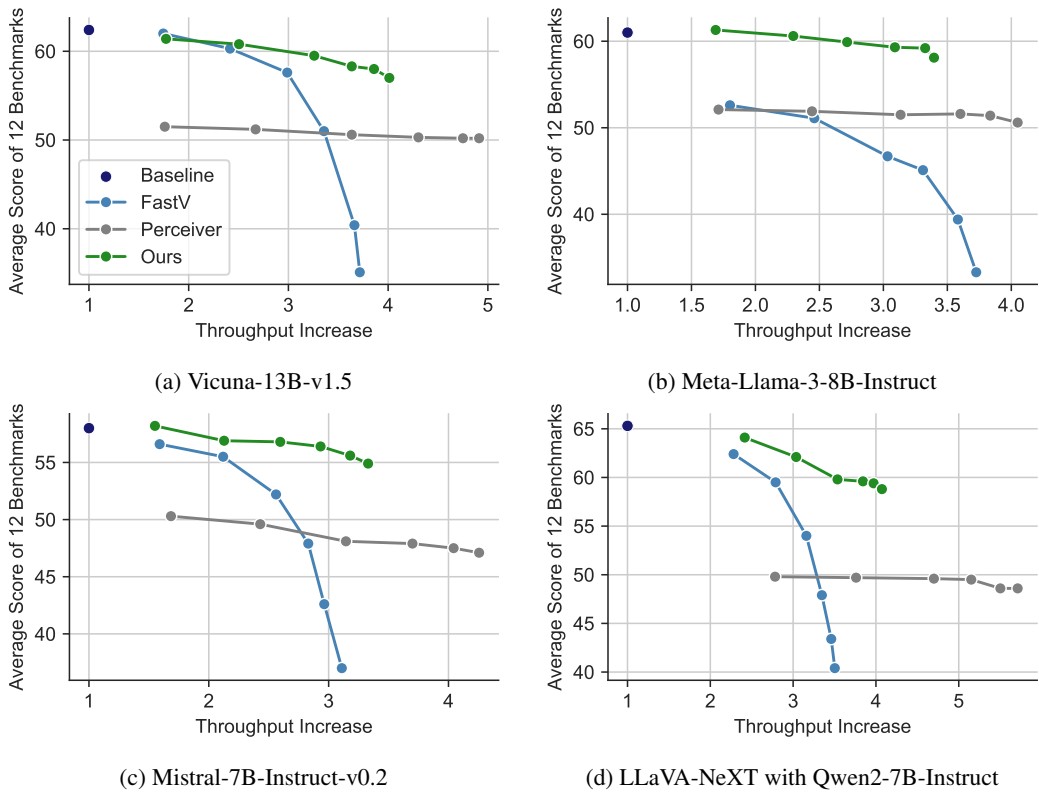

(a) Vicuna-13B-v1.5           (b) Meta-Llama-3-8B-Instruct

(c) Mistral-7B-Instruct-v0.2      (d) LLaVA-NeXT with Qwen2-7B-Instruct

Figure 7: **Efficiency-Performance Trade-Off Curve with Different Language Towers under 2-Token Generation.** Due to the space limit, we show the 128-token generation scenario in Appendix A.4. For the first 3 models, we use 256, 128, 64, 32, 16, and 8 visual tokens to generate the line plot, and for LLaVA-NeXT, we use 512, 256, 128, 64, 32, and 16 visual tokens.

## 5.4 DIFFERENT LANGUAGE TOWERS

We extensively evaluate the effectiveness of our method with different language towers. As shown in Figure 7, replacing the original Vicuna-7B-v1.5 language model with Vicuna-13B-v1.5 (Zheng et al., 2024), Meta-Llama-3-8B-Instruct (Dubey et al., 2024), and Mistral-7B-Instruct-v0.2 (Jiang et al., 2023), Victor remains highly effective and significantly outperforms the two baseline methods. For both Meta-Llama-3-8B-Instruct and Mistral-7B-Instruct-v0.2, Victor demonstrates minimal performance drop and a slow decay in performance as the number of visual tokens decreases. Notably, for these two models, when the number of visual tokens is reduced by half, the method shows no performance degradation at all.

We further demonstrate the performance of our method on a different vision-language model design: LLaVA-NeXT (LLaVA-v1.6) (Liu et al., 2024a). LLaVA-NeXT follows a similar architecture to LLaVA-v1.5 but increases the number of visual tokens from 576 to 2,880 by incorporating different aspect ratios, enhancing the model's capabilities. Additionally, LLaVA-NeXT utilizes Qwen2-7B-Instruct (Yang et al., 2024) as its language tower, benefiting from its extended context length. In our experiments, we reduce the number of visual tokens to 512, 256, 128, 64, 32, and 16. As indicated in Figure 7d, our method remains highly effective in the LLaVA-NeXT setting, consistently outperforming both FastV and the Perceiver Resampler.

## 5.5 DIFFERENT LAYERS TO DROP VISUAL TOKENS

We show the results of the ablation study on which layer to drop the visual tokens (hyperparameter $k$) in Figure 8. In terms of throughput improvement, it is clear that the earlier we drop the visual tokens, the more efficient the model becomes. For lower-layer numbers, such as $k = 1$ or $k = 2$, the model's efficiency significantly increases, with throughput reaching nearly a $4\times$ improvement. However, this comes with a substantial performance drop, suggesting that one or two layers are likely insufficient for the summarization process. In contrast, when $k \geq 3$, the performance degradation is minimal, staying within a $5\%$ performance score loss. Notably, when $k = 5$, with half of the visual tokens dropped, the model experiences no performance loss.

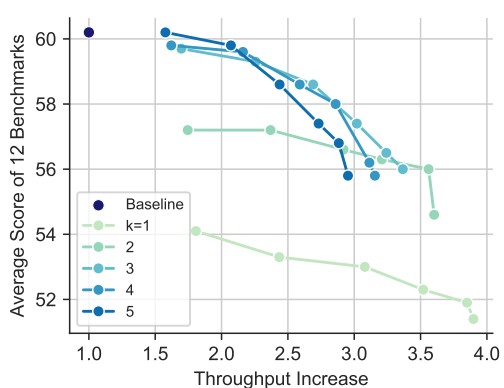

Figure 8: **Ablation on Token Dropping Layers.**

## 5.6 EFFECT OF VISUAL REGISTERS ON REGULAR VLMS

In Figure 9, we present the results of not dropping the visual tokens and instead using visual registers as a means for the model to store useful information, similar to those proposed by Darcet et al. (2023). As reflected in Figure 9, there is a slight performance improvement over the baseline model, but it is limited to around a $2\%$ increase. However, once the number of visual registers exceeds 64, a further increase does not result in additional performance gains. On the other hand, adding more visual tokens leads to a decrease in throughput. Interestingly, adding just 8 visual tokens offers a minimal throughput reduction while still providing a $1\%$ performance boost, making it a "free lunch" for visual-language models.

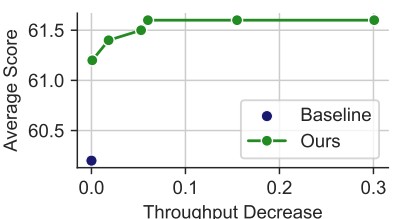

Figure 9: **Results without Dropping Visual Tokens.** From left to right on the line plot, we incrementally add 8, 16, 32, 64, 128, and 256 visual tokens respectively.

## 5.7 ANALYSIS

In Section 3.1, we empirically demonstrate that the visual registers are more compact than the original visual tokens. In this subsection, we perform a simple analysis to examine whether and how visual registers summarize visual information. The attention map from visual registers to visual

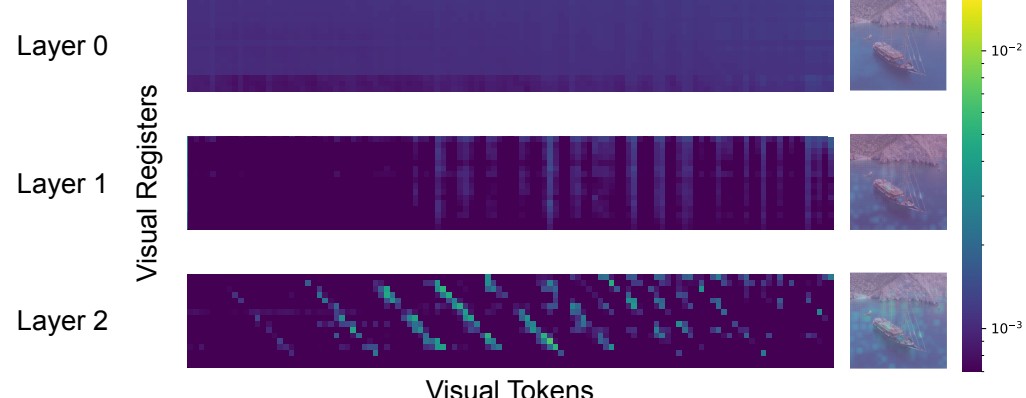

Figure 10: **Attention Map from Visual Registers to Visual Tokens.** We prompt the model with a test image from the COCO dataset and the instruction, "Describe the image."

tokens is shown in Figure 10. Although the model is not explicitly trained to summarize visual information into the visual registers, they implicitly encode the visual tokens, as indicated by the significant attention scores between visual registers and visual tokens. Interestingly, the visual registers exhibit low attention to visual tokens in the first two layers, and the summarization primarily occurs in the third layer, just before the visual tokens are removed. This may be because the first two layers focus on processing the visual tokens or aligning the visual tokens and registers into a shared space to facilitate communication in later layers. This observation aligns with the ablation results discussed in Section 5.5, where dropping visual tokens in the first or second layer causes a significant performance drop. This suggests that it is more effective for the summarization process to occur in the later layers.

As shown on the right side of Figure 10, when examining the attention mapped back to the original image, the visual registers primarily focus on key elements like the rock in the water and the boat mast, while also capturing broader regions of the image. Overall, even without supervision, `Victor` implicitly learns to summarize the image information both effectively and efficiently.

## 6 LIMITATION AND FUTURE WORK

While `Victor` is simple and effective, we identified some limitations and directions for future improvements. Currently, `Victor` is not a training-free method, and it must be incorporated at the training stage of the vision-language modeling. Developing a version of `Victor` that could be applied post-training would be a valuable advancement. However, this might be challenging, as the language tower may need to be specifically trained to learn to effectively utilize the visual registers. Another limitation is the inflexibility of the number of visual registers. As discussed in Appendix A.5, the performance degrades if the number of visual tokens is changed on the fly without retraining. In future work, we believe incorporating certain auxiliary loss functions could help make `Victor` more adaptable and flexible. Additionally, while this paper focuses on applying `Victor` to vision-language models, we believe this technique could also benefit language models, particularly in long-context tasks. We leave these for future exploration.

## 7 CONCLUSION

In this paper, we introduce `Victor`, a novel visual token summarization method that significantly enhances the efficiency-performance trade-off in vision-language models. Without explicit enforcement, the language tower utilizes register tokens to summarize visual information within the first $10\%$ of the layers. After summarization, `Victor` removes the need for visual tokens beyond these layers. Our approach offers a superior balance in efficiency compared to state-of-the-art methods. Moreover, with just up to $0.03\%$ additional parameters, `Victor` is compatible with various attention mechanisms, providing a user-friendly and efficient solution across different hardware environments for future applications.

## REPRODUCIBILITY STATEMENT

As mentioned in Section 4.1, our experiments primarily follow the original LLaVA-v1.5 implementation. Additionally, our method is simple and straightforward to implement. We will release the source code with the camera-ready version.

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

# A APPENDIX

## A.1 LENGTH STATISTICS FOR INDIVIDUAL BENCHMARKS

We show the length statistics of benchmarks in Table 2. Based on the representative lengths of these benchmarks, there are two main categories: 1) short-generation, represented by the 2-token generation scenario in our experiments, and 2) long-generation, represented by the 128-token generation scenario.

Table 2: **Prompt and Generation Length Stats of Individual Benchmarks.**

| | VQAv2 | GQA | ScienceQA | TextVQA | VizWiz | POPE | MME | MMBench | Seed | MMMU | Average |
|---|---|---|---|---|---|---|---|---|---|---|---|
| **Short-Generation Benchmarks** | | | | | | | | | | | |
| Prompt Len. | 43.05 | 46.10 | 93.04 | 43.88 | 44.31 | 43.70 | 54.28 | 86.21 | 93.04 | 210.20 | 75.78 |
| Generation Len. | 1.56 | 1.09 | 2.00 | 8.88 | 3.19 | 1.00 | 1.00 | 1.00 | 2.00 | 1.21 | 2.29 |

| | LLaVA-Bench-Wild | MM-Vet | Average |
|---|---|---|---|
| **Long-Generation Benchmarks** | | | |
| Prompt Len. | 49.82 | 49.74 | 49.78 |
| Generation Len. | 146.60 | 96.11 | 121.36 |

## A.2 PERFORMANCE ON INDIVIDUAL BENCHMARKS

We show the performance on individual benchmarks of Section 5.1 in Figure 11.

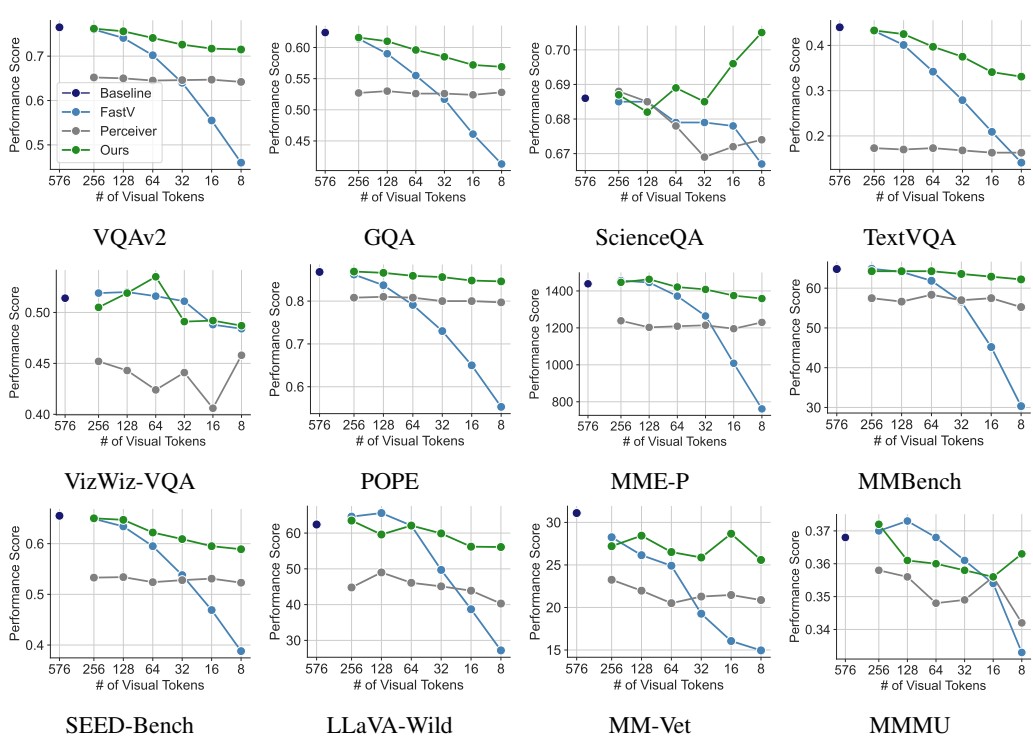

Figure 11: **Individual Benchmark Performance.**

## A.3 TOTAL TRAINING-TIME REDUCTION

The total training-time reduction is shown in Figure 12.

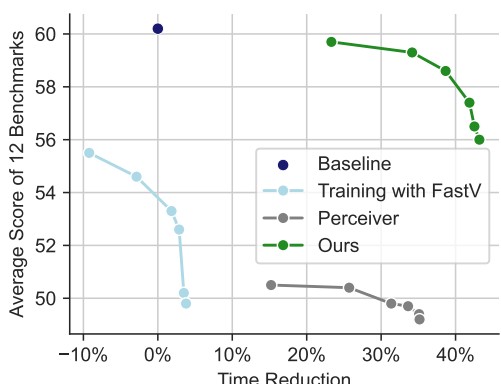

Figure 12: **Performance vs. Total Training-Time Reduction..**

## A.4 EXTRA RESULTS WITH DIFFERENT LANGUAGE TOWERS

The extra result of Section 5.4 with 128-token generation scenario is presented in Figure 13.

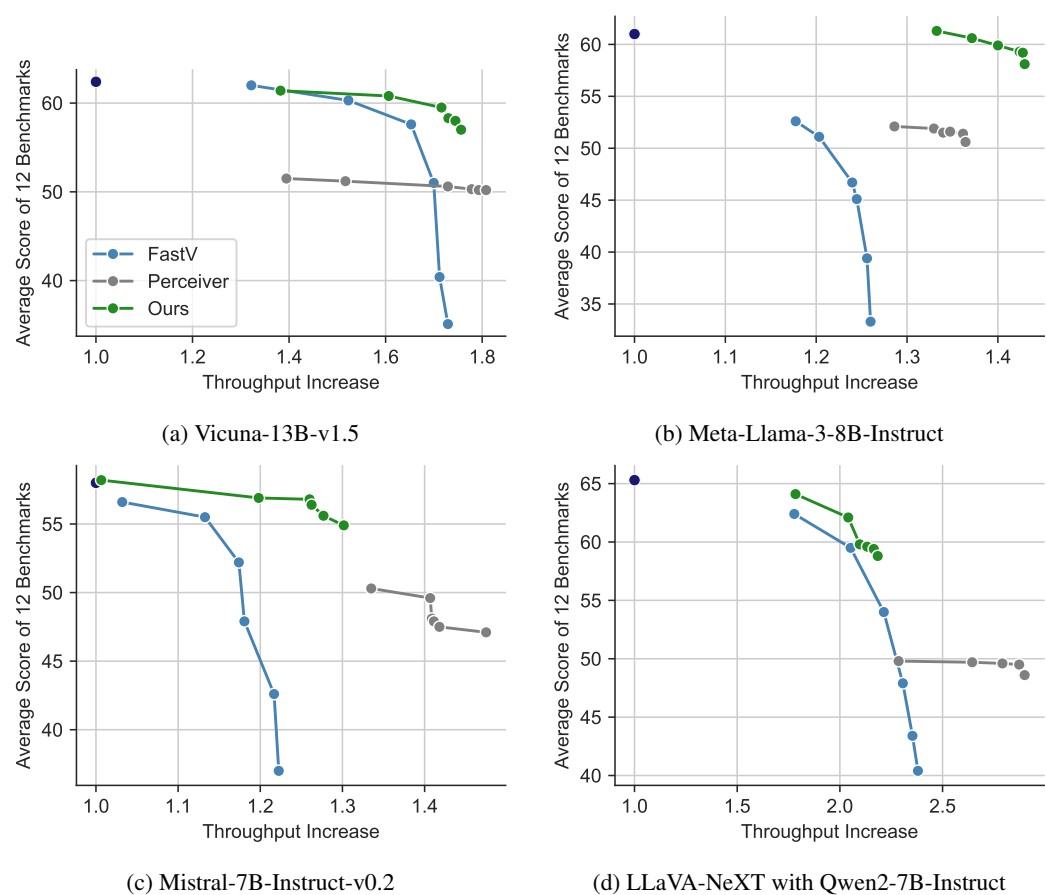

Figure 13: **Efficiency-Performance Trade-Off Curve with Different Language Towers under 128-Token Generation.**

## A.5 Ablation on Adjusting the Number of Visual Registers at Inference

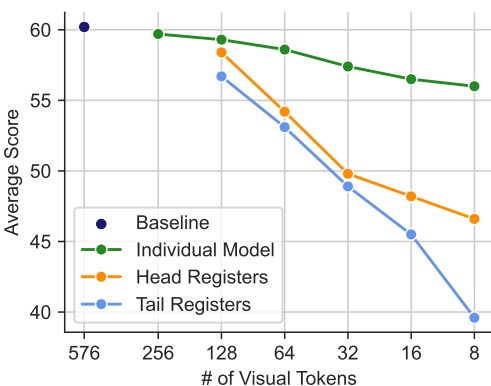

Figure 14: **Ablation on Adjusting the Number of Visual Registers at Inference.**

In our main experiments, we retrain the model whenever a different number of visual registers is required. In this subsection, we explore two strategies for adjusting the number of visual registers dynamically at inference time. Given a `Victor` model with $M$ visual registers, if we want to use $M' < M$ registers, we either select the first $M'$ registers (referred to as "head") or the last $M'$ registers (referred to as "tail"). As shown in Figure 14, the performance of these adjustments is not as effective as retraining the model from scratch. However, we believe that adding certain auxiliary losses during training can make our method more flexible, and we leave this for future work.

## A.6 Importance of Visual Registers for Summarization

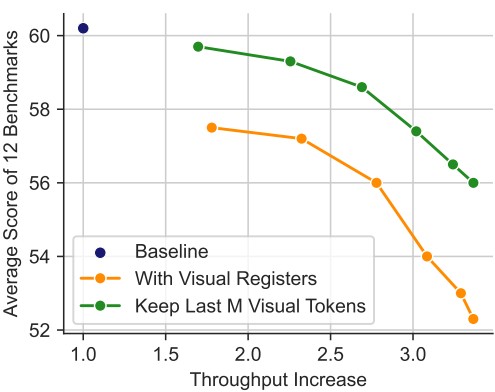

Figure 15: **Importance of Visual Registers for Summarization.**

In this subsection, we conduct an ablation study to demonstrate the necessity of using visual registers for summarization. Specifically, we compare our approach to an alternative method where instead of prepending additional tokens to the visual tokens, we retain the last $M$ visual tokens at layer 3. This requires the model to summarize all visual information into these last existing $M$ visual tokens. As shown in Figure 15, while the ablated method results in a slight improvement in throughput, overall the performance drops significantly. This highlights the importance of incorporating visual registers for effective summarization.

## A.7 DIFFERENT VISUAL REGISTERS

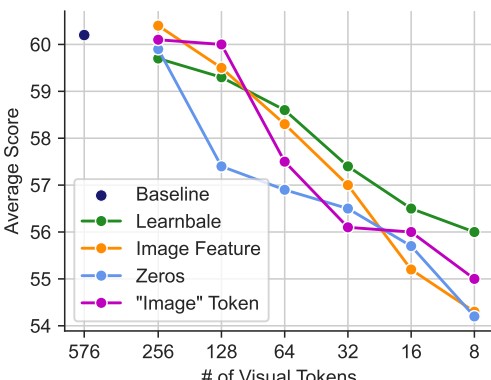

Figure 16: **Ablation with Different Visual Registers.**

We also experiment with various types of visual registers. In addition to using learnable tokens, we test three alternative methods for visual registers: 1) **Pooled Image Feature**: utilizing average-pooled visual tokens as the register tokens, 2) **Zeros**: initializing with all zeros, and 3) **"Image" Token**: using the embedding of the word "Image." The results are presented in Figure 16. The "Image" Token method is effective for the visual registers, especially when the number of visual tokens is reduced to 256 and 128, as there is no performance drop. However, all alternative methods showed relatively worse performance compared to learnable queries in the low visual token regime. Therefore, we adopt learnable queries for `Victor` as they offer better overall performance.

