# OpenReview forum: "Efficient Vision-Language Models by Summarizing Visual Tokens into Compact Registers"
_ICLR.cc/2025/Conference — ICLR 2025 Conference Withdrawn Submission_

### Official Review · Reviewer_t1gv · 2024-10-29

**Soundness:** 3
**Presentation:** 2
**Contribution:** 2
**Rating:** 3
**Confidence:** 4

**Summary:**

The paper introduces Visual Compact Token Registers (Victor), a method to enhance computational efficiency in vision-language models like LLaVA by reducing the large number of visual tokens. Victor replaces these tokens with a small set of “register tokens,” which summarize the visual information within the first few layers. Afterward, the original visual tokens are discarded, minimizing computational load. This approach maintains performance with minimal parameter addition, achieving only a 4% performance drop while cutting training time by 43% and increasing inference speed by 3.36×.

**Strengths:**

1. The paper introduces Visual Compact Token Registers (Victor), a method to enhance computational efficiency in vision-language models like LLaVA by reducing the large number of visual tokens.
2. This work continues to explore the performance of registers in VLM, which is a valuable perspective.
3. The paper is well-written and clearly presents the proposed framework.

**Weaknesses:**

1. This paper repeatedly emphasizes that the main limitation of FastV lies in its reliance on attention maps, which restricts it to using only the simplest attention implementation. However, this is fundamentally an engineering issue related to code implementation and does not merit extensive discussion in the paper. There are two straightforward solutions to address this: (1) set only the second layer’s attention to “eager” mode while employing flash_attention_2 for the remaining layers, or (2) store the key and value states of the second layer and recompute the attention map with (Lt, Lv) shape. Both methods can significantly reduce latency and flops. Therefore, all data related to FastV in the experiments should be reconsidered, particularly the comparison of training times in Section 5.3, where setting all FastV layers to eager mode is highly unreasonable, making the training time reported for FastV invalid.
2. When conducting research on efficiency, the primary focus should be on performance. Ideally, the efficiency strategies we aim to achieve should provide the greatest possible acceleration while maintaining comparable performance levels. However, this paper lacks any tables presenting the specific benchmark scores, relying solely on line graphs, which makes it difficult to assess the exact performance degradation. Moreover, the latter half of the line graphs has limited practical significance; for instance, when the benchmark performance drops by around 4%, the actual performance decreases far more, rendering such acceleration strategies unusable. It would be beneficial to select a few settings and display the key benchmark scores, such as for TextVQA, GQA, and SEED-Bench.
3. When comparing inference acceleration, this paper uses FastV as a major baseline. However, FastV is a training-free method, whereas the approach  proposed by this paper requires complete training process before inference acceleration is possible. Therefore, this comparison is unreasonable from a cost perspective.
4. The reported benchmark performance of fastv after training was even lower than that of inference-only. Based on experimental experience, this result appears highly unreasonable, suggesting a review of the code for potential issues, and there is reason to suspect the baseline may have been artificially lowered.
5. Please provide a simple baseline, just pooling after the first layers to retain visual tokens. According to the paper’s methodology, retained visual tokens are intended to aggregate information from other image tokens. If using learnable tokens is feasible, a simple and straight approach would be to directly leverage part of the original image tokens to gather information.
6. In Section 4.1, it states, "We use the AdamW optimizer (Loshchilov & Hutter, 2017) with a learning rate of 0.0001 and no weight decay for one epoch." I noticed that this learning rate differs from the original LLaVA's rate of 0.001, indicating that you have adjusted the training hyperparameters. If this method were to be applied to another model, it would also require extensive hyperparameter tuning, which could incur significant computational costs.

**Questions:**

1. In Section 5.6, the paper claims that retaining visual tokens can boost performance by 2%, please specify the benchmark and score in which this phenomenon is observed. It is quite unusual, because even in the original register paper[1], retaining visual tokens did not yield notable gains in visual comprehension tasks such as ImageNet.
2. It is worth noting that in Appendix A.2, the baseline score in the MME-P line graph appears to be only around 1,400, which does not align with the 1,510 score reported in the original LLaVA paper, could you explain it?

[1]Vision Transformers Need Registers, https://arxiv.org/abs/2309.16588

---

### Official Review · Reviewer_f1tW · 2024-11-03

**Soundness:** 2
**Presentation:** 3
**Contribution:** 2
**Rating:** 5
**Confidence:** 4

**Summary:**

The paper introduces Visual Compact Token Registers, a novel method aimed at improving the efficiency of vision-language models by reducing the number of visual tokens processed. Victor’s approach involves adding a few learnable register tokens after the initial visual tokens, then using only the first few layers of the language model to summarize the visual information into these registers. After these layers, the model discards all original visual tokens, substantially decreasing computational demands for both training and inference. This method requires minimal new parameters and speed up the training and inference.

Overall, the paper is well-written and the contributions are clear and sound.

**Strengths:**

The paper is well-written. The proposed method Victor is simple yet effective. The authors conducted numerous experiments to clearly show the superior performance of Victor, showcasing its ability to reduce computational costs without significantly compromising model accuracy. The experimental results are presented in a clear and compelling manner.

**Weaknesses:**

1) The paper aims at improving the efficiency of the VLMs. I think more baselines should be incorporated to demonstrate the effectiveness of the method.
Some examples are:
[1] PyramidKV: Dynamic KV Cache Compression based on Pyramidal Information Funneling
[2] LOOK-M: Look-Once Optimization in KV Cache for Efficient Multimodal Long-Context Inference
[3] Efficient Inference of Vision Instruction-Following Models with Elastic Cache

2) Also, I think more methods should be discussed in related work section, about the development of efficient VLMs, currently still only fastV is mentioned.

3) Some questions in the experiments.
    [1] why using 'zero' or 'image' can still maintain the performance to some extend as shown in Figure 16. I do not really get why zero can do something.
    [2] I notice that some benchmarks, such as scienceqa, even perform much better with only 8 visual tokens. Can you provide some explanations?

Due to the questions above, I think some revisions are needed, including more baseline comparisons, revision of the related work section and more explanations on experiment results. Still, I will consider raising my score if my questions are well-solved.

**Questions:**

As stated in the weakness section

---

### Official Review · Reviewer_ukDY · 2024-11-04

**Soundness:** 3
**Presentation:** 3
**Contribution:** 2
**Rating:** 6
**Confidence:** 3

**Summary:**

Motivated by observations from FastV, the authors utilize only the initial transformer layers to summarize the visual tokens into register tokens. Following this summarization, the original visual tokens are discarded, enabling more efficient processing while minimizing information loss. This method can be applied during both the training and inference stages and demonstrates a relatively small performance drop when utilizing fewer visual tokens.

**Strengths:**

1. The proposed method is simple and effective.
2. The authors conduct experiments on various LLMs, aiming to demonstrate the generalizability of their method across different models.

**Weaknesses:**

1. Since the advantages of additional register are not adequately presented. It's hard to attribute the superiority of the experimental results to the proposed method.
2. It is advisable to add a baseline experiment that directly uses fewer visual tokens. For instance, a pooling layer could be applied directly after the image tokenizer to achieve a similar computation FLOPs.

**Questions:**

1. Could you provide a more detailed explanation for the poor training performance observed with FastV in Figure 6?
2. Additionally, according to Figure 16, the learnable registers appear to show significant differences from the image features only when the number of visual tokens is very low (≤ 16). The necessity and advantages of the additional register design need to be demonstrated through more ablation experiments.
3. More broadly, just out of curiosity, can this method be generalized to multiple images?

---

### Official Review · Reviewer_mD9s · 2024-11-04

**Soundness:** 4
**Presentation:** 4
**Contribution:** 3
**Rating:** 8
**Confidence:** 4

**Summary:**

This paper introduces a novel method named Visual Compact Token Registers (Victor) aimed at improving the computational efficiency of vision-language models (VLMs). Based on the finding of [1], Victor minimizes computational overhead by summarizing extensive visual tokens into a smaller set of register tokens that distill the global information from input visual tokens. This process occurs within the initial k layers of the language tower, after which the original visual tokens are discarded to increase the overall throughput of the VLM. Victor is shown to reduce training time by 43% and enhance inference throughput by 3.36×. It only introduce extra 1.78M parameters and even when there is only 8 visual register tokens, it can still maintaining less than a 4% drop in performance.



[1] Darcet, T., Oquab, M., Mairal, J., & Bojanowski, P. (2024). Vision Transformers Need Registers. ICLR.

**Strengths:**

1. **Efficiency Gains**: The method significantly reduces both training time and inference computational requirements without heavily impacting performance. The reported 43% training time reduction and 3.36× throughput boost are compelling.
2. **Implementation Simplicity**: Victor introduces only 1.78M additional parameters, representing 0.03% of the total model size, which is minimal compared to alternatives.
3. **Compatibility with Attention Mechanisms**: Unlike FastV, Victor is compatible with efficient attention implementations such as FlashAttention. This further improve its compatibility and throughput during implementation.
4. **Empirical Validation**: The paper provides extensive experimental evidence across 12 benchmarks, comparing Victor's performance to that of FastV and the Perceiver Resampler. It also provide insightful analysis of how visual registers summarize visual information, the results are reasonable and also align with the experiment discovery.
5. **Flexibility Across Models**: Demonstrated effectiveness with different language towers and vision-language model configurations, showcasing the method’s adaptability.

Overall, This paper presents a solid, well-validated approach to reducing computational overhead in vision-language models with register tokens. The idea is novel and solid, experiment results well support the design choices and also provide valuable insights.

**Weaknesses:**

This paper’s weaknesses are mainly discussed in Sec. 6

1. **Dependence on Training Integration**: The method is not training-free, which may limit its use for pre-trained models that cannot be modified or retrained.
2. **Limited Adaptability of Visual Registers**: The performance drops when the number of visual registers is adjusted post-training. The paper acknowledges this and suggests potential auxiliary loss functions as future work but does not address it in the current method.

Despite these weaknesses, the paper's contributions remain impactful, offering a compelling method that balances efficiency and performance effectively.

**Questions:**

**Extend to Other VLM Architectures:**
Can Victor be extended to other types of vision-language models, such as those using cross-attention mechanisms or early-fusion architectures like MoMA or Chameleon? Clarification on how adaptable the method is across different architectures would provide more insight into it.

---

### Official Review · Reviewer_rAED · 2024-11-05

**Soundness:** 2
**Presentation:** 2
**Contribution:** 2
**Rating:** 3
**Confidence:** 4

**Summary:**

This paper introduces Victor (Visual Compact Token Registers), a method for reducing the number of visual tokens in vision-language models to improve efficiency without significant performance loss. By summarizing visual tokens into compact registers during the initial layers of a language model, Victor reduces computational overhead, achieving substantial reductions in training time and improvements in inference throughput. Victor’s approach is compatible with efficient attention mechanisms and minimizes parameter increase.

**Strengths:**

1. The paper is clearly written and easy to read.
2. The proposed Victor is compatible with high-efficiency attention implementations, such as FlashAttention.

**Weaknesses:**

1. The approach to visual token summarization feels overly simplistic. Prior studies suggest that fewer visual tokens are often sufficient, and Victor’s method lacks a more sophisticated or in-depth design for how register tokens are used to retain critical information. The current version seems to only provide a simple and straightforward baseline, while an exploration of more refined register token strategies could potentially minimize performance drops further or increase acceleration gains, maybe reducing >43% training time. For example, an advanced token-drop strategies, the relationship of register token drop ratios with increasing layer depth should be explored further.
2. Some comparisons with visual token compression models are missed. Victor’s efficiency claims would be more compelling if compared against models that perform early token compression before inputting to the LLM, such as LLaMA-VID. LLaMA-VID uses only two visual tokens to achieve core functionality, this paradigm seems to be also or even more efficient in theory.
3. While Victor is evaluated using different LLM architectures, it relies on only the LLaVA architecture for vision-language fusion. Comparisons with alternative VLM architectures, such as Mini-Gemini, Cambrian-1, or InternVL, would have offered a more robust evaluation.

**Questions:**

See the weaknesses.

---

### Note · Authors · 2024-11-14

I have read and agree with the venue's withdrawal policy on behalf of myself and my co-authors.